# Ten-valley excitonic complexes in charge-tunable monolayer WSe$_2$

Alain Dijkstra [1,2,3,8] ✉, Amine Ben Mhenni [1,2,3,8] ✉, Dinh Van Tuan [4], Elif Çetiner [1,2,3], Muriel Schur-Wilkens[1,2,3], Junghwan Kim [4], Laurin Steiner [1,2,3], Kenji Watanabe [5], Takashi Taniguchi [6], Matteo Barbone [1,2,3], Nathan P. Wilson[1,2,3], Hanan Dery[4,7] & Jonathan J. Finley [1,2,3] ✉

Excitons dominate the optical response of two-dimensional (2D) semiconductors. Strong interactions produce peculiar excitonic complexes, which provide a testing ground for exciton and quantum many-body theories. Here, we report a hitherto unobserved many-body exciton that emerges upon filling both the K and Q valleys of WSe$_2$. We optically probe the exciton landscape using charge-tunable devices with unusually thin dielectrics that facilitate doping up to several $10^{13}$ cm$^{-2}$. We observe the emergence of the thermodynamically stable complex when 10 valleys are electrostatically filled. We gain insight into its physics using magneto-optical measurements. Our results are well-described by a model where the number of distinguishable Fermi seas interacting with the photoexcited electron-hole pair defines the complex's behavior. In addition to expanding the repertoire of excitons in 2D semiconductors, this complex could probe the limit of exciton models and answer open questions about screened Coulomb interactions in 2D semiconductors.

Layered semiconductors and their heterostructures host a plethora of tightly bound exciton complexes, which define enhanced light-matter couplings[1–3]. In monolayer transition metal dichalcogenides (TMDs), the range of excitonic states observed includes neutral excitons, or bound electron-hole pairs[4,5], their excited states[6], charged excitons[7], and biexcitons[8]. Under specific conditions, even six- and eight-particle complexes can be observed[9,10]. In stacked bilayers, this landscape is further enriched by the possibility of forming dipolar and quadrupolar interlayer excitons[11–13] in addition to the emergence of moiré excitons under suitable conditions[1,3].

Though each complex manifests peculiarities, these excitons typically obey elegant valley-contrasting physics due to the broken inversion symmetry. Moreover, they have some properties in common, such as large binding energies, reaching hundreds of meV in the case of neutral excitons[2,14]. The large binding energy stems from quantum confinement effects due to lowered dimensionality and enhanced Coulomb interactions, reflecting weaker screening from the environment, where dynamical screening effects can play a key role[15]. Excitons provide an ideal testbed for studying Coulomb interactions and many-body physics in two dimensions. Excitons also find applications in quantum photonics, for instance, in the realization of emergent bosonic phases, such as excitonic insulators and exciton condensates[2,16]. Furthermore, they have become a useful probe for charge order in strongly correlated systems, such as Mott insulators[17] and Wigner crystals[18].

In functional devices, TMDs can be embedded in nanoscale capacitor structures, allowing for the precise tuning of charge density

---

[1]Walter Schottky Institute, Technical University of Munich, Garching, Germany. [2]Physics Department, TUM School of Natural Sciences, Technical University of Munich, Garching, Germany. [3]Munich Center for Quantum Science and Technology (MCQST), München, Germany. [4]Department of Electrical and Computer Engineering, University of Rochester, Rochester, NY, USA. [5]Research Center for Electronic and Optical Materials, National Institute for Materials Science, Tsukuba, Japan. [6]Research Center for Materials Nanoarchitectonics, National Institute for Materials Science, Tsukuba, Japan. [7]Department of Physics and Astronomy, University of Rochester, Rochester, NY, USA. [8]These authors contributed equally: Alain Dijkstra, Amine Ben Mhenni. ✉e-mail: Alain.Dijkstra@tum.de; Amine.Ben-Mhenni@tum.de; JJ.Finley@tum.de

over a wide range that is significantly larger than in bulk materials. Here, hexagonal boron nitride (hBN), a layered insulator that can be exfoliated down to the monolayer limit and has unique dielectric properties, is often used as the gate dielectric. The ability to seamlessly integrate 2D materials into tunable devices drives physics discoveries using this material platform[1–3].

Here, we identify a novel multiparticle exciton ($M$), upon filling the Q/Q' valleys in WSe$_2$. We study the emergence of these states as a function of carrier density, magnetic ($B$) field, and lattice temperature. Our results show that they arise from the interaction of photoexcited electron-hole pairs with nine distinguishable electron reservoirs, possibly resulting in a 20-particle correlated state. We also discuss how the excitonic response of the system is modified as different Fermi reservoirs are created by electrostatic doping. Good agreement is obtained between experimental data and theoretical predictions.

## Results

### Accessing WSe$_2$ exciton landscape at high carrier density

The device consists of a dual-gated WSe$_2$ monolayer that features 5.3 nm and 6.3 nm thick hBN flakes as gate dielectrics (see "Methods" for fabrication details). This choice is justified by the higher breakdown electric field for thinner hBN flakes. The breakdown field of hBN tends to be around 0.6 V nm$^{-1}$ for typical hBN gate dielectric thicknesses (10–20 nm)[19,20] and can be as high as 2–3 V nm$^{-1}$ in the limit of few-layer flakes[21]. In our device, each of the hBN dielectrics is capable of withstanding fields greater than 1 V nm$^{-1}$ while maintaining a leakage current below 5 nA. As such, the large voltages that can be applied are found to be capable of reaching electrostatic doping levels larger than $2 \times 10^{13}$ cm$^{-2}$.

Figure 1a shows gate-dependent reflection contrast spectra of the device. We apply the same gate voltage ($V_G$) to both the top and bottom gates relative to the WSe$_2$ monolayer. The neutral exciton and its first excited state are the prominent features around charge neutrality (resonances $X_{1s}^0$ and $X_{2s}^0$ at $V_G$ -0 V). The Fermi level can be tuned into the valence band (VB) by applying a negative $V_G$, injecting holes into the WSe$_2$, and promoting the formation of the positively charged exciton ($X^+$). Here, the precise optimization of the dielectric stack for the low-energy excitonic states allows tracking of the blueshifting and weakening of the $X^+$ resonance over nearly 200 meV as the hole density is increased.

Conversely, a positive $V_G$ injects electrons into the WSe$_2$ monolayer, giving rise to a variety of negatively charged excitons. Figure 1b

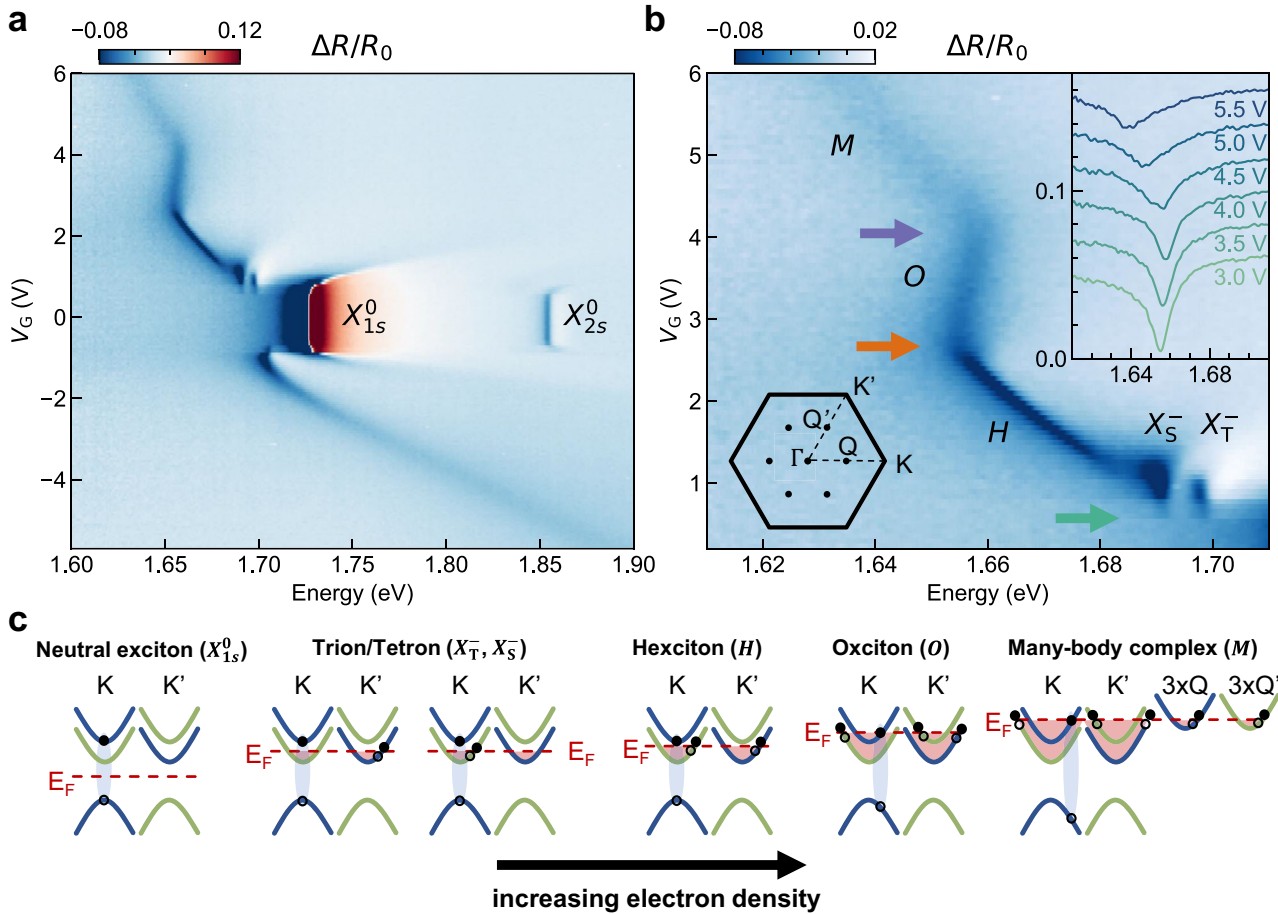

**Fig. 1 | Device structure and gate-dependent optical response of WSe$_2$. a** Gate-dependent reflection contrast spectra of a WSe$_2$ monolayer taken at 4 K showing neutral (around $V_G$ -0 V), negatively charged (positive $V_G$), and positively charged excitons (negative $V_G$). **b** Close-up view of the negatively charged regime from (**a**) revealing a variety of excitonic complexes: the exchange-split singlet and triplet trions ($X_{S,T}^-$), the hexciton ($H$), the oxciton ($O$), and another many-body exciton ($M$). Arrows mark the onset of the filling of the lower conduction band (CB) valleys at K/K' (green), the upper CB valleys at K/K' (orange), and the lower CB valleys at Q/Q' (purple). Inset top right: reflection contrast spectra from the same dataset over the transition from $O$ to $M$. Inset bottom left: the Brillouin zone of the 2D WSe$_2$ crystal with labels to show the $\Gamma$ point, the K/K' points and the Q/Q' points right in between the latter two. **c** Band diagram schematics of the neutral exciton and negatively charged excitons for increasing electron doping. Initially, electrons start filling the lower spin-orbit split K/K' valleys as the density increases, sequentially promoting the formation of singlet and triplet trion/tetron-, hexciton-, and oxciton complexes. Eventually, the Fermi level reaches the Q/Q' valley band edge, giving rise to an even larger $M$ exciton involving charges from both the K/K' and Q/Q' valleys. The complexes are depicted as the binding between the photoexcited e-h pair and Fermi particle-hole excitations of the distinguishable Fermi seas, wherein the CB holes of the Fermi seas move together and are correlated with the complex.

shows a close-up view of these resonances. The richer excitonic physics on the electron doped side can be traced back to the smaller spin-orbit splitting in the conduction band (CB), $\Delta_c = 12$ meV[22,23], compared to $\Delta_v = 400$ meV in the VB[24]. At each critical density, the oscillator strength is transferred from smaller to progressively larger many-body excitonic states involving larger numbers of correlated particles. This behavior is accompanied by the observation of a change in the energy shift rate with respect to the electron density. As the electron density increases, the reflection contrast spectra subsequently show the exchange-split singlet and triplet negative trions ($X_{S,T}^-$)[25,26], hexcitons ($H$)[9,10,27], and oxcitons ($O$)[9]. Remarkably, the oscillator strength transitions from $O$ to an even larger and previously unreported many-body exciton ($M$). This is shown by an abrupt switching from the slight blueshifting of $O$ to a marked redshifting behavior. The inset in Fig. 1b shows extracted reflection contrast spectra of the transition from $O$ to $M$. We confirmed the generality of this observation by studying a second WSe$_2$ device with 17 nm and 28 nm thick hBN gate dielectrics. This sample yields an identical transition from $O$ to $M$ (Supplementary Data Fig. 1).

To determine the origin and composition of the many-body exciton $M$, we continue to discuss the band diagram of WSe$_2$, the sequential filling of the CB valleys, and the ensuing decay and emergence of excitonic complexes. In our approach, we rely on the composite excitonic states model[9,10,27,28], motivated by its successful description of the $H$ and $O$ excitons. This model takes the void left behind by a bound Fermi electron—or the *Fermi hole*—into consideration, and correspondingly treats each electron that is bound to a photoexcited electron-hole pair as a *Fermi particle-hole pair*. As such, $X^0$ is a photoexcited electron-hole pair that is not bound to a Fermi particle-hole pair (first panel of Fig. 1c), while the charged excitons are photoexcited electron-hole pairs bound to Fermi particle-hole pairs from one or more Fermi reservoirs (remaining panels of Fig. 1c), depending on the doping level.

Building upon this model, we classify a photoexcited electron-hole pair by its *distinguishability* and an excitonic complex via its *optimal* or *suboptimal* character[28]. A photoexcited electron-hole pair is distinguishable only if both charges reside in valleys without Fermi reservoirs. Only in this case do the charges of the photoexcited electron-hole pair have unique quantum numbers that are not shared with carriers in the Fermi sea. An excitonic complex is optimal if it involves a Fermi particle-hole excitation from each available Fermi reservoir and suboptimal otherwise. Importantly, all bound Fermi particle-hole pairs have to be distinguishable in at least one quantum number, i.e., they must have unique valley and/or spin, with respect to each other and to the photoexcited electron-hole pair. The optimal/suboptimal character of an excitonic complex and the indistinguishability of its photoexcited electron-hole pair are key to understanding its gate-dependent energy shift and broadening. For the sake of completeness, we start by discussing $X_{S,T}^-$, $H$, and $O$, respectively, and then finish by extending this framework to $M$.

$X_{S,T}^-$ forms upon filling of the lower (CB) at the K/K' points. The filling onset is indicated by the green arrow in Fig. 1b. The second panel of Fig. 1c shows schematics of $X_{S,T}^-$. While $X_{S,T}^-$ are distinguishable, they are suboptimal because there is an available Fermi sea that is not contributing a Fermi electron-hole pair.

At a sufficiently large electron density in the lower CB, the photoexcited electron-hole pair interacts with two Fermi seas—one from the K and one from the K' valley—giving rise to the six-particle complex $H$ (third panel of Fig. 1c). In contrast to $X_{S,T}^-$, not only is the photoexcited electron-hole pair distinguishable, but $H$ is optimal since it involves a Fermi electron-hole pair from each available Fermi reservoir. $H$ experiences an energy redshift, and does not broaden or decay with increasing electron density. These observations are both hallmarks for optimal and distinguishable excitonic complexes[28]. This redshifting behavior originates from the interplay of the bandgap renormalization

(BGR) and the binding energy reduction due to increased screening of the Fermi sea electrons $\Delta E(n_T) = \Delta E_g(n_T) - \Delta E_b(n_T)$, where $n_T$ is the total electron density. The binding energy reduction $\Delta E_b$ being smaller than the bandgap reduction $\Delta E_g$, a net redshift $\Delta E$ remains[15,29–32].

$V_G$ -2.6 V marks the filling onset of the upper CBs at the K/K' valleys (orange arrow in Fig. 1b). At this point, an additional Fermi reservoir becomes available, and the photoexcited electron-hole pair now binds to three distinguishable Fermi electron-hole pairs, giving rise to the eight-body complex $O$ (fourth panel of Fig. 1c). While $O$ is an optimal complex, it is also indistinguishable because the electron of the photoexcited electron-hole pair resides in a valley that contains a Fermi sea. The transition from $H$ to $O$ is marked by a switch from redshifting to blueshifting behavior. The excitation of the photo-excited electron into the already filled upper K valley requires the resident electrons to spatially rearrange to satisfy the Pauli exclusion principle. This process is known as a shakeup, and causes the photo-excitation of an electron-hole pair to require higher energy with rising Fermi level[28]. This contribution adds to the BGR and binding energy reduction, yielding a net energy blueshift. The shakeup also causes the resonance to broaden with increasing charge density, confirmed by dispersive Lorentzian fits of the reflection contrast spectra shown in Supplementary data Fig. 2.

Finally, $V_G$ -4.2 V (purple arrow in Fig. 1b) marks yet another transition—this time from $O$ to $M$, with a change from blueshifting to redshifting behavior and a sudden increase in width. This transition does not lead to a loss in oscillator strength, which remains almost constant for the $H$, $O$, and $M$ resonances (see Supplementary data Fig. 2). Analogous to previous transitions, an additional set of Fermi particle-hole excitations must be involved in the formation of $M$. Since the lower and upper K/K' valleys each already provide a Fermi particle-hole pair to form $O$, the additional Fermi particle-hole excitations must stem from Fermi reservoirs residing in different valleys.

In the following section, we will show that the additional pairs stem from the three-fold degenerate Q/Q' valleys. This means that complex $M$ consists of a photoexcited electron-hole pair, which binds up to 9 other Fermi particle-hole pairs from distinguishable Fermi seas and involves the correlated interaction of up to 20 quasiparticles (rightmost panel in Fig. 1c).

## Q valley electrons contribution
We proceed by elucidating the origin of $M$. Hereby, we first link the precise charge density to $V_G$, using magneto-optic experiments (see Methods). This calibration matches between $V_G$ on the left axis of Fig. 2a and the charge densities on the right axis of this figure.

Furthermore, we have performed valley population calculations to obtain the evolution of the electron density in each valley as a function of the total electron density ($n_T$), as shown in the right panel of Fig. 2a. Finally, by self-consistently comparing the experimental data with the calculations, we assign the transitions—from $H$ to $O$ and from $O$ to $M$—to the filling onset of the corresponding valleys.

The valley population calculations rely on the minimization of the total energy of the electron gas for each total electron density $n_T$. In describing the total energy, we account for the kinetic energy due to filling of the valleys plus the exchange energy among electrons that share the same spin-valley configuration (see "Methods" for details). For our calculations, we use a CB spin-orbit splitting ($\Delta_c$) of 12 meV based on ref. 22. This predicts that the upper CB valleys at K/K' start to be populated for $n_T$ -0.7 × 10$^{13}$ cm$^{-2}$. This value is in good agreement with the experimental value of 0.8 × 10$^{13}$ cm$^{-2}$ extracted from the reflection contrast data. Note that accounting for the exchange interaction is of paramount importance: Only considering the valley filling (kinetic energy) yields less than half the density than experimentally observed. Furthermore, in order to determine the energy separation between the lower CB valleys at K/K' and the lower CB valleys at Q/Q' ($\Delta_{KQ}$), we use the experimentally determined critical electron density

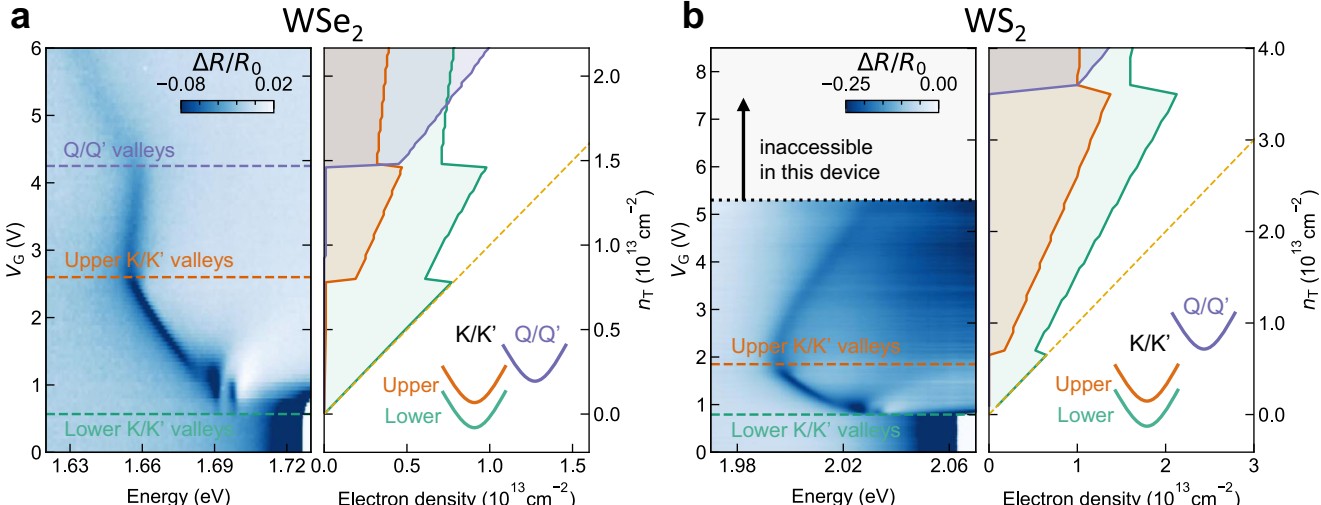

**Fig. 2 | Filling of the Q valley and comparison with WS₂. a** The electron-doped side of the 4 K gate-dependent reflection contrast measurement of WSe$_2$ is shown on the left side in which horizontal dashed lines mark the onset of the filling of the lower K/K′ valleys (green), upper K/K′ valleys (orange) and lower Q/Q′ valleys (purple). Abutting this data is a calculation of the distribution of carriers in the different CB valleys as a function of the overall carrier density by minimizing the total energy of the electron gas (kinetic and exchange). The charge density scales are matched by calibrating the reflection contrast data using magneto-optic experiments (see "Methods"). The onsets of the resonances *O* and *M* are perfectly reproduced by using $\Delta_c = 12$ meV for the spin-orbit splitting in the K/K′ valleys[22,23], and $\Delta_{KQ} = 30$ meV for the energy difference between the lower CB valleys of K and Q. **b** The same as in (**a**) but for a WS$_2$ sample with comparable hBN dielectric

thicknesses to those in the main WSe$_2$ sample (5.3 nm and 3 nm for the top and bottom dielectric, respectively). To reach the maximum density while avoiding breakdown, we apply $V_G$ to the top gate and 0.8 $V_G$ to the bottom gate. Horizontal dashed lines mark the onset of the filling of the lower and upper K/K′ valleys in green and orange, respectively. The right-hand side shows a calculation of the distribution of charges in the different CB valleys in the WS$_2$ monolayer. The gate-voltage-dependent charge density of the reflection contrast data was determined based on the calibration of the WSe$_2$ sample in (**a**), and adapted using a simple capacitor model (see "Methods" for details). The filling of the K/K′ valleys shows great similarity with WSe$_2$, but contrastingly the filling of the Q/Q′ valleys would only happen at the experimentally inaccessible electron density of ~$4 \times 10^{13}$ cm$^{-2}$ because $\Delta_{KQ} = 81$ meV in a WS$_2$ monolayer[24].

at which the Q/Q′ valleys start filling ($n_{KQ}$) of $1.5 \times 10^{13}$ cm$^{-2}$. Using an iterative strategy, we find excellent agreement between experiment and theory for $\Delta_{KQ} = 30$ meV (see Fig. 4 of the Supplementary Data), in agreement with $\Delta_{KQ} = 35$ meV, estimated from first-principles calculations[24].

The calculations provide an important insight into the red-shifting behavior of *M*. As soon as the Q/Q′ valleys begin to fill, the electron density in the K/K′ valleys remains almost constant. This means that starting from that point, the majority of additionally injected charges populate the Q/Q′ valleys. This can be explained by the three-fold degeneracy, higher density of states stemming from the larger effective mass, and larger contribution from exchange energy. Combined, these properties result in a slower rise of both the Fermi level and the electron density in the upper K/K′ with increasing $n_T$. Therefore, shakeup processes, of which the blueshifting contribution is determined by the increasing electron density in the upper K/K′ valleys, are mitigated. On the contrary, the redshifting contribution from the screening-induced interplay between BGR and reduction in the binding energy remains, since it depends on $n_T$ directly.

We explain the formation of *M* by putting the extracted electron densities into perspective. *M* first emerges at a density of $1.5 \times 10^{13}$ cm$^{-2}$, corresponding to ~10 free electrons in every 8 × 8 nm$^2$ of the monolayer. These 10 electrons are accommodated in the least restrictive way if they all have different quantum numbers, dictated by the Pauli exclusion principle. They are thus distributed over the lower- and upper K/K′ valleys and the threefold degenerate Q/Q′ valleys. When a photoexcited electron-hole pair is introduced, it binds to these distinguishable electrons. The alternative is to scatter the electrons away, which will deplete the charge in the region of the complex, creating an energetically unfavorable charge inhomogeneity in the monolayer.

We continue to show that a WS$_2$ device, having access to a similar charge density range, also exhibits *H* and *O* as a consequence of the

similar $\Delta_c$. However, the *M* exciton is not observed since the Q/Q′ valleys are energetically farther away.

WS$_2$ and WSe$_2$ monolayers are similar in terms of $\Delta_c$, $g$-factors, and the types of excitonic complexes and their binding energies[10,22,33–35]. These parameters are related to the band character of the K/K′ valleys, which are mostly formed from transition metal W orbitals, the shared element between WS$_2$ and WSe$_2$. In contrast, the Q/Q′ valleys have additional contributions from the $p$ orbitals of the chalcogenide S/Se orbitals. We expect this to result in a noticeable difference in $\Delta_{KQ}$ between the two materials.

The left panel of Fig. 2b shows the gate-dependent reflection contrast spectra of the WS$_2$ device. Here, we calibrated the electron density based on parameters obtained from our study of the WSe$_2$ device (see "Methods"). We notice a clear similarity between the two material systems, when looking at the successive emergence of the neutral exciton ($X^0$), the trions ($X_{S,T}^-$), the appearance of the hexciton *H* and oxciton *O*, and all of their corresponding binding energies. Another similarity is that the filling of the upper CB valleys at K/K′ in WS$_2$ manifests at a similar carrier density as in WSe$_2$: ~$0.6 \times 10^{13}$ cm$^{-2}$ versus ~$0.8 \times 10^{13}$ cm$^{-2}$, respectively, marked by orange dashed lines.

Contrasting with these similarities, we do not observe filling of the Q/Q′ valleys in the WS$_2$ sample, despite reaching a higher maximum carrier density of ~$2.3 \times 10^{13}$ cm$^{-2}$ compared to the ~$2.2 \times 10^{13}$ cm$^{-2}$ in the WSe$_2$ sample. This observation is explained by the valley population calculation of WS$_2$, in the right panel of Fig. 2b. Employing $\Delta_{KQ} = 81$ meV[24], predicts that a carrier density of ~$3.6 \times 10^{13}$ cm$^{-2}$ would be required to start populating the Q/Q′ valleys. Our experiments not only verify that such large electron density is out of our experimental range, but also confirm that $\Delta_{KQ}$ is critically different in WSe$_2$ and WS$_2$ monolayers. This understanding reinforces our claim that the emergence of *M* in WSe$_2$ is associated with filling of the CB valleys at Q/Q′.

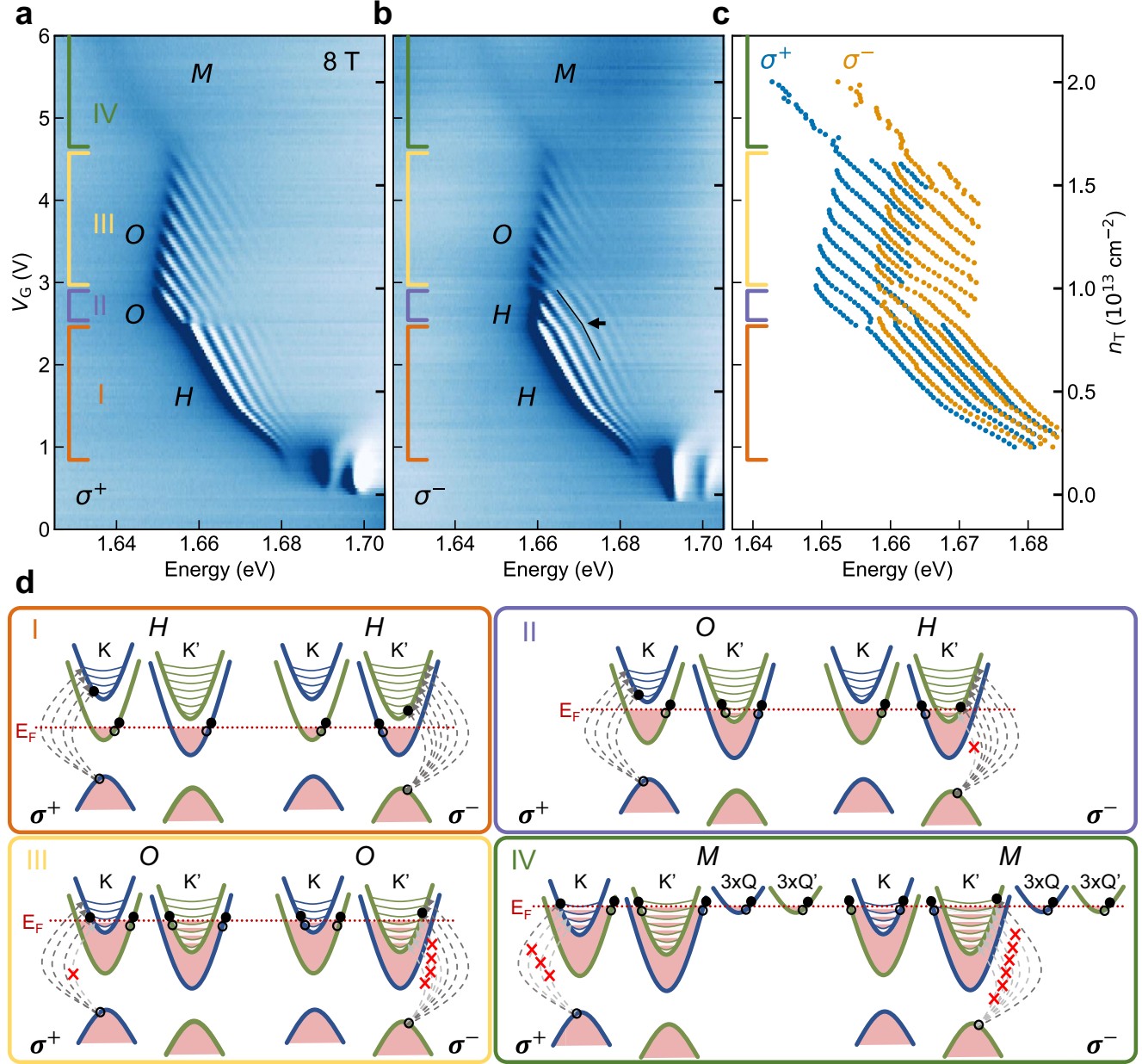

**Fig. 3 | Magneto-optics of WSe₂.** Gate-dependent reflection contrast measurements performed at 8 T magnetic field and a mixing chamber temperature of ~20 mK, resolved for right-handed circularly polarized light $\sigma^+$ (**a**) and left-handed circularly polarized light $\sigma^-$ (**b**). Four gating regimes are marked with colored brackets and are labeled I–IV, which sequentially show $H, O, O,$ and $M$ excitons for the $\sigma^+$ measurement and $H, H, O,$ and $M$ excitons for the $\sigma^-$ measurement. **c** A plot of the four energetically lowest, resolvable, local minima, per spectrum, extracted from (**a**) and (**b**) using a peak-detection method described in the "Methods". Left (right) ticks in (**a–c**) correspond to the voltage (density) scale on the far left (right) of the figure. **d** Bandstructure models explaining the observed resonances in (**a, b**) for the different gating regimes I–IV. In **I**, the lower K– and K' valleys are filled and hexcitons form for both $\sigma^+$ and $\sigma^-$. The additionally observed resonances in (**a, b**) are due to Landau levels in the upper K– and K' valleys. In **II**, the upper K' valley is partially filled and hosts a Fermi sea. Therefore, $O$ is observed for $\sigma^+$ yet $H$ is still observed for $\sigma^-$, now with a different slope than in **I**, indicated with a black arrow in (**b**). As the Landau levels in the upper K' valley fill, resonances disappear for $\sigma^-$ due to Pauli blocking. In **III**, all CB valleys at K and K' are populated, both $\sigma^+$ and $\sigma^-$ show $O$ excitons, and the systematic disappearance of resonances due to the filling of Landau levels. In **IV**, in addition to the K/K' valleys, the Q/Q' valleys are populated, allowing for the formation of $M$. No Landau level reminiscent oscillations are observed for $M$. We note that Landau quantization also occurs in the VB and the lower CB valleys at K and K'[36,38,45,46]. Regardless, they do not affect the spectroscopic features we study in this work at a magnetic field of 8 T; therefore, we have chosen not to include them in our bandstructure drawings shown in (**c**) to improve clarity.

## Magneto-optical study

To gain more insight into the nature of $M$, we perform gate-dependent polarization resolved, reflection contrast measurements at a magnetic field of 8 T. Typical results are presented in Fig. 3a, b for right- and left-handed circularly polarized light, respectively. By resolving the helicity, we address the K and K' valleys independently,

which is essential because the spin and valley Zeeman splitting breaks their degeneracy.

We identify four gating regimes, marked with colored brackets in Fig. 3a, b. To overlay the gate-dependent reflection contrast spectra of both polarizations, the local minima of the reflection contrast data were determined (see "Methods") and are plotted in Fig. 3c. For each of

the four regimes, a corresponding band structure sketch is shown in one of the boxes of the corresponding color in Fig. 3d.

Starting with regime I, both $\sigma^+$ and $\sigma^-$ show a fan of equally spaced $H$ exciton resonances, reminiscent of Landau levels[36–38], unlike the $X^-_{S,T}$ excitons which exhibit a single resonance each. In the structure of a $X^-_{S,T}$ exciton, the relative motion of the hole is nearly twice that of the two electrons. This allows the hole to spend equal times near each of the two electrons, thereby creating a tightly bound trion[39]. This results in the center-of-mass motion being Landau quantized, but not the internal dynamics of its constituent particles.

In contrast to the internal structures of $X^-_{S,T}$, $H$ (and $O$) is composed of a dark trion in its core and satellite electron(s) from the optically-active top valleys[9,27]. The relative motion of the optically active satellite electron is slower than that of the three particles in the core trion. Therefore, the photoexcitation of $H$ resembles the Landau level quantized motion of a free electron[9].

We corroborate this interpretation by extracting an effective mass from the Landau levels and comparing it to that of free electrons in the upper CB. We start from the energetic spacing of the resonances in regime I ($\Delta_{LL}$), which is constant for a given voltage but increases slightly from 3.2 to 3.6 meV from the bottom to the top of regime I and is identical for both polarizations. Calculating the corresponding cyclotron frequency yields an effective mass in the range $0.26–0.29m_0$ in the single particle approximation, in units of the free-electron mass $m_0$. This value closely matches the $0.29m_0$ effective mass of the upper conduction band. Moreover, it does not match the exciton reduced mass, given by $m_e m_h/(m_e + m_h) = 0.16m_0$[6]. Therefore, the observed resonances are due to the electron of the photoexcited electron-hole pair being excited to different Landau levels in the upper CB (see panel I in Fig. 3d) rather than the Landau levels of an exciton. The slight change of the energetic spacing of the resonances might be due to an enhancement in binding of the photoexcited electron to the complex[9].

Regime II starts with the filling of the upper K' CB valley (see panel II in Fig. 3d). For the $\sigma^+$ polarization, an additional distinguishable Fermi sea becomes available, and $O$ excitons are formed instead of $H$. This is accompanied by a ~2 meV redshift of the full set of the $\sigma^+$ resonances due to the binding energy of the additional electron, in agreement with previous observations[36,37]. At the same time, for the $\sigma^-$ polarization, the lowest-energy resonance disappears. This is caused by the filling of the first Landau level in the upper K' CB, which makes it unavailable for the photoexcited electron-hole pair due to Pauli blocking. The remaining $\sigma^-$ resonances in regime II are thus still $H$. A striking feature of the transition from regime I to II in Fig. 3b is the change in slope of the $H$ resonances, marked with a black arrow, which signals a change in the rate at which the Fermi level rises under the injection of carriers. This change in slope is a compelling argument that an additional band is being filled.

The third regime commences when the lowest energy resonance vanishes in the $\sigma^+$ data due to Pauli blocking (see panel III in Fig. 3d), which marks the first Landau level in the upper K CB valley being filled. For the $\sigma^-$ polarization, this transition manifests as a small redshift of the resonances (see Fig. 3b), as they transition from $H$ to $O$. Each spectrum for both polarizations shows periodic Landau level related resonances of $\Delta_{LL} = 3.6$ meV over the full regime, matching the effective mass of the upper CB at K/K'.

For increasing gate voltage, resonances disappear in a periodic fashion, where each disappearance marks an additional Landau level in the upper CB at K having been filled. This provides a path to calibrating the total charge density as a function of the applied gate voltage, where the ratio of the effective masses between the upper and lower CBs at K/K' is the only parameter (see "Methods").

At $V_G$ ~4.7 V, the filling of the Q/Q' valleys and the appearance of $M$ denote regime IV. We note that, determined by the g-factor of the Q/Q' valleys, these valleys will show a Zeeman splitting and lose their degeneracy. Depending on this shift, it is possible that the Q' valley populates before the Q valley, and the many-body complex would then only interact with three additional Fermi reservoirs rather than six. This means that the observed complex under a strong magnetic field would consist of up to 14 particles rather than 20. However, we estimate the g-factor in the Q valleys to be around $g_Q$ ~1[34] and therefore do not expect a selective population of the Q/Q' valleys at 8 T.

Strikingly, the Landau level resonances disappear upon filling the Q/Q' valleys, and $M$ emerges as a single, broadened feature for both polarizations. We speculate that the availability of additional reservoirs (i.e., the Q/Q' valleys) enhances the pure dephasing via scattering processes between K and Q, broadening the individual contributions to the resonance to an extent that they are irresolvable. This explanation is supported by magneto-optic photoluminescence (PL) experiments (see Supplementary Data Fig. 5), where Landau level resonances stemming from the upper K valley are observed for the $M$ exciton. Crucially, we observe only resonances for Landau levels that lie energetically below the Q/Q' valley onset, where scattering is unfavorable, and no resonances for Landau levels above the Q/Q' valley onset. Nonetheless, further investigations are required to understand intervalley scattering and its impact on the optical response.

## Thermodynamic stability

We continue by studying the thermodynamic stability of the newly observed multi-particle exciton $M$. Figure 4 shows gate-dependent reflection contrast measurements at increasing temperatures. For the lowest temperature (20 K) the characteristic changes in energy shift with respect to electron density at ~2.8 V and at ~4.5 V, corresponding to the formation of the $O$ and the $M$ resonances respectively, are maintained. At 30 K, the transition from $O$ to $M$ is barely discernible, only a change in width and intensity of the resonance being observed starting from ~4 V. When the temperature is further increased to 40 K, $O$ and $M$ can no longer be distinguished. These observations suggest that the ionization energy of the many-body complex $M$ lies in the range of $k_B T$ ~2.5–3.5 meV.

To corroborate the thermodynamic stability of $M$, we perform gate-dependent photoluminescence experiments on the same sample. Typical results are presented in Fig. 4b. Starting from ~0.1 V, $X^0_{1s}$ start to quench and the $X^-_{S,T}$ excitons gain in oscillator strength marking that the lower K/K' valleys start to fill (green arrow). At ~0.9 V the brighter $H$ emission takes over, which does not broaden with increasing voltage. The filling of the upper K/K' valleys (orange arrow) is evidenced by an asymmetric broadening and a double peak visible near the maximum intensity, which we attribute to the oscillator strength transitioning from $H$ to $O$. At ~4.6 V (purple arrow), we observe the filling of the Q/Q' valleys, as a distinct change in the rate of redshifting, and a change in broadening of the emission peak.

The $X^0_{1s}$-, $X^-_{S,T}$- and $H$ excitons, as observed in reflection contrast (see Fig. 1), energetically closely follow the corresponding photoluminescence peaks in Fig. 4b. A different behavior is observed for the $O$- and $M$ excitons, where the maxima of the photoluminescence peaks show a redshift with respect to their corresponding reflection contrast resonances. We understand this different behavior because the $O$- and $M$ excitons are complexes with an indistinguishable photoexcited electron-hole pair. This means that for the excitation of a photoexcited electron-hole pair, a shakeup of the Fermi sea is required to respect the Pauli exclusion principle, adding a blueshifting term. For emission, there is no Pauli blocking, and therefore, this term is lacking. This is evidenced by the almost identical slope that the $H$-, $O$-, and $M$ excitons show in photoluminescence, whereas clear kinks are observed in reflection contrast. Instead, the high-energy side of the broadened $O$- and $M$ photoluminescence peaks closely follow their reflection contrast counterparts, which we interpret as emission in which the photoexcited electron resides at the Fermi level. This clear overlap is another compelling argument that the observed emission is

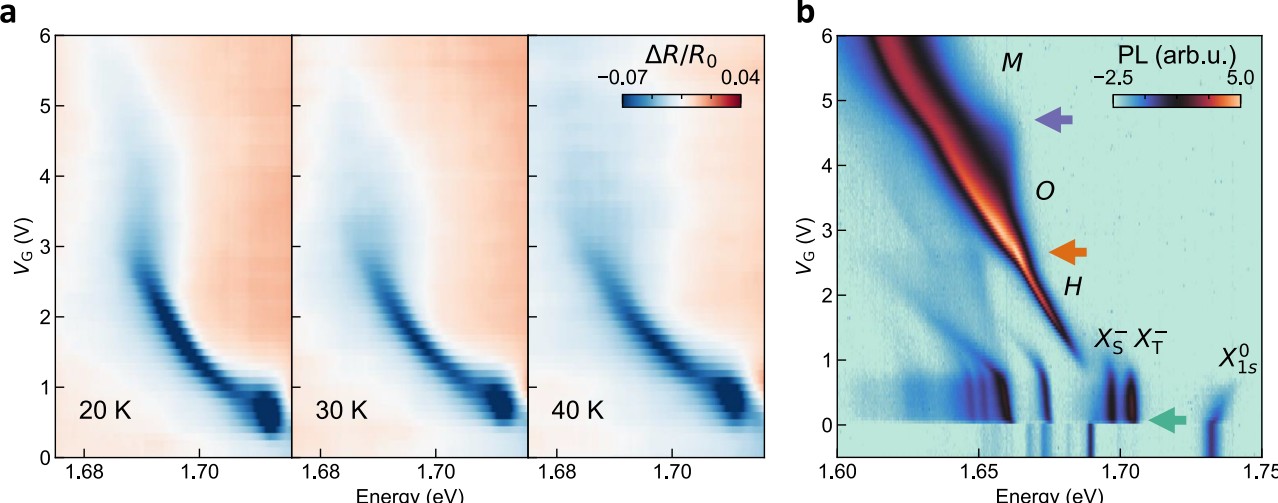

**Fig. 4 | Stability of the many-body complex. a** Gate-dependent reflection contrast measurements at increasing temperatures, in the range of 20–40 K. With increasing temperature, the clear redshift of the many-body complex ($M$) fades and becomes indistinguishable for a temperature in between 30 and 40 K, corresponding to an ionization energy of -3 meV for $M$. **b** Gate-dependent photoluminescence (PL) measurement taken at 4 K, plotted on a logarithmic scale. At $V_G = 0$ V, we observe $X_{1s}^0$, indicating charge-neutrality. Starting from $V_G$ -0.1 V, $X_{1s}^0$ quenches and the $X_{S,T}^-$ excitons emerge, marking that the lower K/K' valleys start to fill (green arrow). This onset happens at a lower $V_G$ than for the reflection contrast measurement in Fig. 1a, due to additional charge carriers introduced into the system by the excitation laser. At -0.9 V the oscillator strength is shifted from the $X_{S,T}^-$ excitons to the $H$ exciton. The filling of the upper K/K' valleys starts at -2.6 V (orange arrow), accompanied by a transition from $H$ to $O$. At -4.6 V (purple arrow), the filling of the Q/Q' valleys is signaled with the onset of a sudden increased rate of redshift of the maximum of the PL signal and a change in broadening.

due to the $O^-$ or $M$ excitons rather than other, lower energy, complexes. Thus, both $O$ and $M$ are not only correlated states with the highest oscillator strength and therefore visible in reflection contrast at their corresponding carrier densities, but they also appear to be sufficiently stable to participate in radiative transitions.

## Discussion

We have observed a ten-valley excitonic complex that emerges upon filling the Q/Q' valleys in monolayer WSe$_2$. The contribution of electrons from the three-fold degenerate Q/Q' valleys, in addition to those from the K/K' valleys, means that these states could involve as many as 20 quasiparticles. Insights from the composite excitonic states model elucidated the nature of exciton $M$. Theoretical follow-up studies include predicting these states' exact composition, binding energy, and decay/recombination pathways. The validity of the various exciton models could be assessed by checking these predictions against the thermodynamic stability and PL presented here. The 2D nature of these states, combined with the large number of particles originating at different valleys, is likely to give rise to complex dynamics. Additional insight into this strongly correlated exciton-Fermi sea system could be gained via ultrafast time-resolved techniques and diffusion studies. Finally, the recipe used in this work could be readily applied to study exotic phenomena occurring at high charge density regimes in materials, with moiré potentials, with engineered broken mirror symmetry[40], and those with inherent magnetic properties[41].

## Methods
### Sample preparation
TMD, graphite, and hBN flakes were prepared by mechanically exfoliating bulk crystals onto substrates with a 70 nm thick layer of SiO$_2$. Flakes were chosen based on criteria including optical contrast, morphology, and surface cleanliness. The sample was constructed using a two-step dry transfer process involving polycarbonate (PC) films. Flakes were picked up at a temperature of 120 °C. Subsequently, the assembled stack, still on the stamp, was cleaned by repeatedly pressing it against the substrate at 155 °C, a process that physically expels trapped bubbles from between the layers. To dissolve the polymer, the sample was first immersed in chloroform for 30 min and afterwards in isopropanol (IPA) for 10 min. The transfer matrix method for plane wave propagation was employed to determine the exact thickness of the overall dielectric structure for optimal optical contrast, after which an additional hBN flake was added on top of the stack. Electrical contacts were defined via standard maskless optical lithography, followed by electron beam evaporation of Cr/Au electrodes with thicknesses of 5 and 100 nm.

### Optical spectroscopy
All optical measurements (with the exception of the magnetic field dependent data) are performed in a confocal microscope setup fitted to an Attodry 800 closed cycle cryostat with a base temperature of -3.8 K and accurate temperature control. A tungsten bulb is used as a white light source to measure reflection contrast. The emitted light is coupled through a single-mode fiber, subsequently collimated and focused to a spot of -2 μm on the sample using a 40 × apochromatic objective. The collected signal is sent through a spatial filter and coupled into an Andor spectrometer equipped with an open diode CCD camera to suppress fringing. The photoluminescence experiments are performed using the same setup and configuration as the reflection contrast measurements, but now, a Helium-Neon laser is used as an excitation source that is focused down to a diffraction-limited spot. For the measurement shown in Fig. 4b, a laser power of -350 nW is used on the sample. A 650 nm long pass filter is used to filter out the laser before the photoluminescence signal is coupled into the spectrometer. The magnetic field-dependent measurements are performed in a Bluefors dilution refrigerator fitted with free space optics forming a confocal microscope in back reflection geometry. The mixing chamber flange is at a temperature of -20 mK when the data is collected. A strongly attenuated NKT SuperK EVO supercontinuum laser is used as an unpolarized white light source and is focused down to a -1 μm spot onto the sample using a low-temperature apochromatic objective for the reflection contrast measurements shown in Fig. 3. The back reflected light is coupled through a quarter waveplate and a linear polarizer to select for left- or right-handed circularly polarized light. The magneto-photoluminescence data shown in

Supplementary Data Fig. 5 are collected in the same setup. A Helium-Neon laser is coupled through a linear polarizer and a quarter wave-plate to excite with circularly polarized light; the optics for the back-reflected light remain the same.

## Data treatment

Reflection contrast is defined as $\frac{R-R_0}{R_0}$, in which $R$ is a gate-voltage dependent signal measured on a site of the sample where the TMD, top gate, and bottom gate are present. $R_0$ is a reference signal, collected as close as possible to the measurement spot, where all layers in the stacked sample are present except for the TMD. Etaloning fringes due to the detector are subsequently removed by creating a voltage-averaged background in which only regions of the two-dimensional dataset are taken into account that do not contain any spectral fea-tures, and subtracting this background from the dataset. The indivi-dually plotted reflection contrast spectra in the inset of Fig. 1b, extracted from the two-dimensional dataset, have been smoothed by convoluting the data with a boxcar function.

In Fig. 3c, the local minima of the reflection contrast spectra in Fig. 3a, b are plotted, which are determined using a peak detection scheme. First, a light Gaussian filter is applied to the original reflection contrast data to reduce noise after which the *find_peaks* function of the *SciPy* package of the Python language is used on $\log\left(-\frac{R-R_0}{R_0}\right)$. Then, all detected peaks outside of the region of interest are rejected. Finally, to enhance the resolution and accuracy of the method, the final energy of each detected peak is determined by fitting the direct vicinity (up to ±4 pixels in the energy direction) of the detected peak with a Lor-entzian and taking its center energy. For the plot in Fig. 3c, the four detected minima for each voltage, with the lowest energy, are included.

Each photoluminescence (PL) signal is measured three times sequentially. After subtraction of the dark counts, the three identical spectra are used to run a statistical outlier detection script that flags and removes the cosmic rays from the spectra.

## Reflection contrast fitting

The reflection contrast data presented in Fig. 1 are fitted using a dis-persive Lorentzian function[18], of which the results are given in Sup-plementary Data Fig. 2. The function,

$$
\begin{aligned}
R_c(E) = &A\cos(\phi)\frac{\gamma/2}{(E-E_0)^2+\gamma^2/4} \\
&+ A\sin(\phi)\frac{E_0-E}{(E-E_0)^2+\gamma^2/4} + C
\end{aligned}
\tag{1}
$$

describes a Lorentzian response of the TMD while compensating for dispersive effects of the dielectric environment. Here, $A$ is the ampli-tude, $E_0$ is the center energy, $\gamma$ is the full width at half maximum, $C$ is an overall offset, and $\phi$ is a phase that describes the dispersion. For the fitted data, we only use spectra that show a single resonance (either the $H$, $O$, or $M$ exciton) to omit fitting several dispersive Lorentzians simultaneously. To remove errors caused by small drifts in the inten-sity or spectrum of the light bulb used for the measurement, we average the signal in between 1.58 and 1.6 eV per spectrum and sub-tract this value from the full spectrum, this way, each spectrum has the same baseline, and we can set $C=0$. Then, a fit is performed for each spectrum, with data used in the vicinity of the resonance, where we allow the $A$, $E_0$, and $\gamma$ to fit freely and $\phi$ to slowly change with voltage.

## Valley population calculations

The charge distribution among the CB valleys is calculated by mini-mizing the total energy of the electron gas at zero temperature[42]. Taking into account the spin-split K/K′ valleys and the lower six valleys

of Q/Q′, the total energy is

$$
\begin{aligned}
E_T = &\frac{\pi\hbar^2}{2}\left[\frac{n_u^2}{m_u}+\frac{n_l^2}{m_l}+\frac{n_Q^2}{3m_Q}\right]+\Delta_c n_u+\Delta_{KQ}n_Q \\
&- \sum_{i=l,u,Q}\frac{C_i}{4\pi^3}\int_0^{k_F^i}dk\,k\int_0^{2k_F^i}dq\,q\,V_{s,q}\int_0^\pi d\varphi\,\theta(|\mathbf{k}-\mathbf{q}|-k_F^i).
\end{aligned}
\tag{2}
$$

The first line corresponds to the total kinetic energy, where $n_{l(u)}$ is the total electron density in the lower (upper) valleys of K and K′, and $n_Q$ is the total electron density in the lower valleys of Q and Q′. $m_i$ is the effective electron mass in the $i$-th valley ($i=\{l, u, Q\}$). The energy splitting between the lower and upper valleys of K/K′ is $\Delta_c$, and $\Delta_{KQ}$ is the corresponding splitting between the lower valleys of K/K′ and the lower valleys of Q/Q′. The second line in Eq. (2) is the contribution due to exchange interactions between indistinguishable carriers (i.e., electrons with similar spin-valley configurations). The Fermi wave-number in the $i$-th valley is $k_F^i=\sqrt{2\pi n_i/C_i}$ where $C_l=C_u=1$ and $C_Q=3$. The integration over the angle $\varphi$ is limited by the Heaviside step function $\theta(\dots)$, where $|\mathbf{k}-\mathbf{q}|=\sqrt{k^2+q^2-2kq\cos\varphi}$. Finally, we have used the static limit of the random phase approximation to describe the screened Coulomb potential at these elevated charge densities,

$$
V_{s,q}=\frac{2\pi e^2}{\epsilon}\cdot\frac{1}{q(1+r_*q)+\kappa_q}.
\tag{3}
$$

$\epsilon$ is the static dielectric constant of hBN, and $r_*=r_0/\epsilon$ where $r_0$ is the polarizability of the monolayer[43]. The screening wavenumber due to electrostatic doping is[42]

$$
\kappa_q=\sum_{i=l,u,Q}\frac{2C_i}{a_i}\left[1-\theta\left(1-\frac{8\pi n_i}{C_i q^2}\right)\sqrt{1-\frac{8\pi n_i}{C_i q^2}}\right],
\tag{4}
$$

where $a_i=\hbar^2\epsilon/e^2 m_i$ is the effective Bohr radius.

The minimization of Eq. (2) involves trying various combinations of valley densities under the constraint of a fixed total density, $n_T=n_u+n_l+n_Q$. The effective masses used in the calculations are $m_l=0.4m_0$, $m_u=0.29m_0$, and $m_Q=\sqrt{0.45\cdot0.75}m_0$ for the WSe$_2$ monolayer, and $m_l=0.36m_0$, $m_u=0.27m_0$, and $m_Q=\sqrt{0.54\cdot0.74}m_0$ for the WS$_2$ monolayer[24]. The effective mass in the Q/Q′ valleys takes into account the mass anisotropy (i.e., the lighter mass along the $\Gamma$-K axis and the heavier mass along the perpendicular direction). In both monolayers, the polarizability parameter is $r_0=4.5$ nm[44], and the K valleys energy spin splittings are $\Delta_c=12$ meV[22]. Finally, $\Delta_{KQ}=81$ meV[24] was used for the WS$_2$ monolayer. We have extracted the value $\Delta_{KQ}=30$ meV for the WSe$_2$ monolayer by matching the measured threshold density in the transition from $O$ to $M$ in Fig. 2a. This extracted value is very close to the first-principles calculated parameter of this monolayer $\Delta_{KQ}=35$ meV[24].

## Carrier density calibration

For the calibration of the carrier density $n_T$ as function of applied gate voltage $V$ of the WSe$_2$ sample, a linear relation between the two is assumed $n_T=a(V-V_0)$, which is justified by a simple double capacitor model. We take the first voltage at which the $X_{2S}^0$ starts to shift as $V_0$, as it is the resonance in our spectrum that is the most sensitive to charges introduced into the TMD. The slope $a$ is calibrated using the Landau levels observed in Fig. 3a. In regime III (labeled in the figure), the disappearance of each resonance marks the filling of a single Landau level in the upper K valley. The number of states of a single Landau level in the upper K valley is given by $n_{uK}=\frac{gB}{2\pi\hbar}$, and the energetic

spacing between the Landau levels by the cyclotron energy $\omega_{uK}$ given by $\hbar\omega_{uK} = \frac{\hbar qB}{m_u}$ in which Q is the electron charge, $B$ the applied magnetic field, and $m_u$ the electron effective mass of the upper CB. One can define an effective density of states $\sigma_{uK} = \frac{\Delta n_{uK}}{\Delta E}$ for the Landau quantized valley considering the number of states $n_{uK}$ for each energy interval $\hbar\omega_{uK}$, yielding $\sigma_{uK} = \frac{m_u}{2\pi\hbar^2}$. Since the magnetic field does not add or remove states, this result is identical to the two-dimensional density of states without a magnetic field.

Although the disappearing resonances are an excellent marker to measure the exact charge density in the upper K valley, at the same time also the upper K' and the lower K and K' valleys are being filled in regime III and the total density of states is thus given by $\sigma_T = \frac{1}{2\pi\hbar^2}(2m_u + 2m_l)$ in which $m_l$ and $m_u$ are the effective masses of the lower and upper conduction bands respectively at the K and K' points.

The slope of the calibration is defined as $a = \frac{\Delta n_T}{\Delta V}$ with $V$ the applied gate voltage. We now choose $\Delta V$ to be the voltage interval between two consecutively disappearing Landau levels and we can therefore use the corresponding energy interval $\hbar\omega_{uK}$ to define $\Delta n_T = \sigma_T \cdot \hbar\omega_{uK} = \sigma_T \cdot \frac{n_{uK}}{\sigma_{uK}}$. Filling out the previous expressions yields $\frac{\Delta n_T}{\Delta V} = \frac{B}{\Delta V}\frac{q}{\pi\hbar}(1 + \frac{m_l}{m_u})$ for the calibration. Here $\frac{B}{\Delta V}$ is accurately determined by measuring the energies of all observed Landau levels at several magnetic fields (see Supplementary Data Fig. 3). Then, the only input parameter of this calibration is $\frac{m_l}{m_u}$ where we use the literature[24] values $m_l = 0.4m_0$ and $m_u = 0.29m_0$, yielding a calibration slope $a$ of $4.0 \times 10^{12}$ cm$^{-2}$ V$^{-1}$.

To approximate the charge density as function of gate voltage for the reflection contrast data of the WS$_2$ sample shown in Fig. 2b, the well-known double capacitor model is used,

$$n_T = \frac{\epsilon_0\epsilon_{hBN}}{q}\left[\frac{1}{d_t}(V_t - V_{t0}) + \frac{1}{d_b}(V_b - V_{b0})\right] \quad (5)$$

$\epsilon_0$ is the vacuum permittivity, $\epsilon_{hBN}$ the effective relative permittivity of the used hBN flakes, $V_t$ and $V_b$ the applied gate voltages to the top and bottom gate, $V_{t0}$ and $V_{b0}$ the gate voltages at which the TMD starts charging for the top and bottom gate, and $d_t$ and $d_b$ the thickness of the top and bottom hBN flake, respectively. In the known literature, a large spread in values for $\epsilon_{hBN}$ is found. In addition, the proportionality between the applied voltage and actual charge density in the TMD seems to be altered for thin hBN layers. This is already exemplified by the difference between the main WSe$_2$ sample with thin hBN layers and the control device with thicker hBN encapsulation (see Supplementary Data Fig. 1). Here, a ~20% difference in $\epsilon_{hBN}$ is necessary to make the $H$ to $O$ transition happen at the same density, despite the fact that the hBN flakes stem from the same batch. We stress that we take note of this observation without making any claims to its physical origin, we use the effective relative permittivity $\epsilon_{hBN}$ merely as a proportionality constant. To find a most reasonable calibration for our WS$_2$ sample, we compare it directly to the WSe$_2$ sample because it exhibits very similar hBN thicknesses and has hBN from the same batch as the WS$_2$ sample. We have calculated an effective $\epsilon_{hBN} = 2.1$ based on the WSe$_2$ sample and applied it to the double capacitor model to find a calibration for the WS$_2$ sample.

## Data availability

The datasets generated and analyzed for this study are available in the mediaTUM repository, https://doi.org/10.14459/2025mp1793118.

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

## Acknowledgements

A.D. acknowledges funding from the European Union's Horizon 2020 research and innovation program under the Marie Skłodowska-Curie (grant agreement No. 101111251). A.B.M. acknowledges funding from the International Max Planck Research School for Quantum Science and Technology (IMPRS-QST doctoral fellowship) and support from the Deutsche Forschungsgemeinschaft (DFG, German Research Foundation) via Germany's Excellence Strategy (MCQST, EXC-2111/390814868). K.W. and T.T. acknowledge support from the JSPS KAKENHI (Grant Numbers 20H00354 and 23H02052) and World Premier International Research Center Initiative (WPI), MEXT, Japan. Work at the University of Rochester was supported by the Department of Energy, Basic Energy Sciences, Division of Materials Sciences and Engineering under Award No. DE-SC0014349. J.J.F. gratefully acknowledges the Deutsche Forschungsgemeinschaft (DFG, German Research Foundation) for financial support (Grant numbers INST95-1642-1, MCQST EXC-2111, e-conversion EXC-2089, and FI 947/7-2).

## Author contributions

A.B.M., A.D., and H.D. conceived the idea. A.D., A.B.M., and J.J.F. managed the project. A.B.M., A.D., and E.C. fabricated the devices. A.D., A.B.M., M.S.W., and E.C. performed the optical measurements. A.D., A.B.M., and L.S. performed the magnetic field-dependent optical experiments. H.D., D.V.T., J.K., and A.D. developed the charge density model and performed the calculations. A.D. and A.B.M. analyzed the results in consultation with H.D., J.J.F., N.P.W., D.V.T., and M.B. K.W. and T.T. grew bulk hBN crystals. A.D., A.B.M., H.D., and J.J.F. prepared the manuscript with input from all authors.

## Funding

## Competing interests

The authors declare no competing interests.
