## [Transparent Peer Review file · Nature Communications]

Ten-valley excitonic complexes in charge-tunable monolayer WSe₂

Corresponding Author: Dr Alain Dijkstra

Version 0:

Reviewer comments:

Reviewer #1

(Remarks to the Author)

The manuscript NCOMMS-25-36475 by Dijkstra et al. reports the observation of a multi-valley coupled excitonic state in a highly electron-doped WSe₂ monolayer. The authors systematically characterize its fundamental properties through doping-, magnetic field-, and temperature-dependent optical spectroscopies. A key finding is the first observation of a new excitonic resonance, referred to as the M-exciton, whose multi-valley nature involving Q/Q' and K/K' valleys is supported by additional measurements and theoretical calculations. The manuscript is well-written and advances our understanding of many-body physics in van der Waals materials.

While the findings are certainly suitable for publication in Nature Communications, I believe the manuscript could be further strengthened by addressing the following concerns:

(1) I generally agree with the experimental observations presented. However, I am concerned about the inconsistency between the dR and PL spectra regarding the Landau level resonances. The manuscript suggests that the presence of additional electronic reservoirs may explain the discrepancy. I think this could be plausible if the Landau level resonances appear in the dR spectra but are absent in PL. However, the opposite scenario is observed here so this casts doubt on the proposed interpretation of the M-exciton. The authors should clarify this inconsistency and provide a more robust explanation.

(2) In the dR measurements, the hexciton, oxciton, and M-exciton exhibit both red and blue shifts, depending on magnetic field or doping, whereas in the PL spectra, these excitons consistently show only red shifts. Could the authors explain the origin of this discrepancy in spectral behavior between dR and PL?

Reviewer #2

(Remarks to the Author)

The authors performed reflectivity and photoluminescence measurements on a gate-tunable device based on WSe₂ (and WS₂) in a previously unexplored range of electron densities. By increasing the carrier concentration they tuned the Fermi level to observe many-body excitations (such as hexcitons and oxcitons) formed by charge carriers populating the higher-energy band with a minimum at the K/K' points. With further increases in electron concentration the authors accessed the regime of many-body complexes (M), which emerge as the Fermi level enters the Q/Q' valleys. In the latter part of the manuscript the authors investigate the properties of these M-complexes using optical spectroscopy performed in an external magnetic field and as a function of sample temperature.

I find this work very interesting and well-written, with a clearly conveyed message. The article is well-structured, and the presentation is appealing. Below, I outline minor comments that the authors should address before publication in Nature Communications:

1. In the Introduction (paragraph starting at line 75), the authors discuss the trion/polaron controversy. However, this paragraph does not seem clearly connected to the rest of the manuscript. The trion/polaron issue is scarcely addressed in the main text, which focuses primarily on H, O, and M excitations. Clarifying the relevance of this discussion to the overall study would be beneficial. Otherwise, I suggest removing said paragraph.

2. In the experimental section (starting at line 97), the temperature at which the measurements were conducted is not reported. Currently, the reader must refer to the Methods section to find this information. It would improve clarity to include the measurement temperature either directly in the main text or in the relevant figure caption.

3. In Figure 3, the tick positions differ between panels (a) and (b), yet panel (b) lacks tick labels. For consistency and readability tick labels should be included in panel (b).
4. Starting at line 424, the authors discuss a change of slope; however, this change is not easily discernible in the presented data. Including a guide to the eye would help the reader follow this discussion more easily.

As mentioned above, the paper is well-written and, in my opinion, complete. However, I would like to offer a few observations that could further improve the manuscript. I emphasize that it is not necessary for the authors to follow any of the suggestions below for successful publication in Nature Communications.

1. Is the oscillator strength (OS) conserved when the O excitation transitions to the M excitation? Have the authors determined the OS from the reflectivity (R) spectra?
2. Line 510 – I believe the strong photoluminescence (PL) observed for the O and M excitations is not surprising, as the PL yield is a product of oscillator strength and occupation. The occupation should be high since O and M correspond to the energetically lowest excitations.
3. The lack of a blueshift for the O transition in PL should be addressed—some hypothesis or discussion would be helpful.
4. Line 466 – Landau levels (LLs) are not visible in reflectivity but are clearly observed in PL for the M excitation. Could the visibility of LLs in PL provide insights into the scattering rate?
5. Line 399 – The increase in the splitting between consecutive LLs is most likely due to band non-parabolicity. A brief discussion or acknowledgment of this point could add value.

Reviewer #3

(Remarks to the Author)

This manuscript by Dijkstra, et al. reports the experimental discovery of a previously unexplored many-body excitonic complex (“M”) in electrostatically doped monolayer WSe₂. Using 5-6 nm thin hBN gate dielectrics to reach exceptional high charge densities, the authors investigate the formation of neutral, charged, and many-body excitonic complexes via reflection contrast and magneto-optical spectroscopy. A key result is the identification of a thermodynamically stable exciton involving interactions with up to 10 filled conduction band valleys, potentially comprising as many as 20 quasiparticles—a remarkable extension of known excitonic states. The finding provides a unique insight into extremal many-body physics in two-dimensional semiconductors and opening pathways for testing the limits of excitonic models under strong doping.

Overall, the data is of high quality, and the result contributes a great new insight to quantum many-body excitonic complexes in 2D materials with broad impact to condensed matter research and quantum physics. The text is well organized and clearly written. I recommend the publication of this manuscript in its present form.

Major Comments:

While the manuscript discusses the “M” complex as involving interactions with up to 9 Fermi particle-hole excitations (yielding up to 20 total particles), it remains somewhat speculative in the absence of direct spectroscopic signatures of individual components. The manuscript would benefit from explicitly addressing this limitation and delineating the extent to which the proposed configuration is supported by experimental observables (e.g., energy shifts, broadening, thermal stability) versus modeled assumptions. More explicit discussion of its competing theoretical frameworks would strengthen the conclusion.

Version 1:

Reviewer comments:

Reviewer #1

(Remarks to the Author)

The authors of NCOMMS-25-36475 by Dijkstra et al. have satisfactorily addressed the points raised during the review process, leading to a significantly improved manuscript. The reported results are important and timely, and I believe they will be of broad interest to the community. In summary, I find the revised manuscript suitable for publication in Nature Communications and recommend its acceptance.

Reviewer #2

(Remarks to the Author)

All my comments and suggestions have been addressed in detail. I recommend the manuscript for publication.

Reviewer #1

The manuscript NCOMMS-25-36475 by Dijkstra et al. reports the observation of a multi-valley coupled excitonic state in a highly electron-doped WSe_2 monolayer. The authors systematically characterize its fundamental properties through doping-, magnetic field-, and temperature-dependent optical spectroscopies. A key finding is the first observation of a new excitonic resonance, referred to as the M-exciton, whose multi-valley nature involving Q/Q' and K/K' valleys is supported by additional measurements and theoretical calculations. **The manuscript is well-written and advances our understanding of many-body physics in van der Waals materials. While the findings are certainly suitable for publication in Nature Communications,** I believe the manuscript could be further strengthened by addressing the following concerns:

We thank the Reviewer for their supportive comments and for their interest in the relevance of our work and its implications. Moreover, we thank the Reviewer for their strong support for the publication of our manuscript.

The referee raises an excellent point. They pointed out that we should provide a discussion that reconciles the differences in our PL and dR data, both in terms of Landau level resonances and to the energy shift. This led us to critically reevaluate our PL data related to the Landau levels, which improved our overall understanding, and by extension, the manuscript.

The answers to both questions are related, and we present them in reversed order.

(2) In the dR measurements, the hexciton, oxciton, and M-exciton exhibit both red and blue shifts, depending on magnetic field or doping, whereas in the PL spectra, these excitons consistently show only red shifts. Could the authors explain the origin of this discrepancy in spectral behavior between dR and PL?

We thank Referees 1 and 2 for raising this comment. We indeed do not comment on the discrepancy between the redshifting behavior of the RC and PL data, but this is a very valid point to understand the *H*, *O*, and *M* excitons.

There are two contributions to the resonance energy shift of an optimal excitonic complex when the charge density increases. The first contribution is a screening-induced energy redshift, which is a universal effect caused by a stronger shrinkage of the band gap energy than the reduction in binding energy. The shrinkage causes a redshift that is slightly stronger than the blueshift caused by the reduced binding energy, and the overall net effect is an energy redshift that is commensurate with charge density (this is described in the manuscript starting from line 197). When the photoexcited electron-hole pair of the excitonic complex is distinguishable (e.g., hexciton in WSe_2), the screening-induced effect is the only contribution, and we get a similar energy redshift in both PL and dR.

The second contribution to the energy shift of the resonance happens only when the photoexcited electron-hole pair is indistinguishable (in WSe_2 this concerns the *O* and the *M* excitons). The photoexcitation of an indistinguishable photoexcited electron-hole pair requires additional energy to overcome the Pauli blocking of already residing excitons in the conduction band. This process is therefore accompanied by a Burstein-Moss or shakeup process. A shakeup means that electrons from the same valley and spin of the photoexcited electron are scattered out of the region of the complex. The required photon energy $\hbar\omega$ equals the energy of the created optimal complex (E_{oc}) plus the energy to overcome the Pauli exclusion (δ), such that $\hbar\omega = E_{oc} + \delta$. Here, δ is commensurate with the charge density. Thus, we find that for dR, resonances with an indistinguishable electron-hole pair

(i.e. the *O*- and *M* excitons) have an additional blueshifting term δ . (This second contribution is described starting from line 214 of the main text.)

What we did not include in the text is that for recombination processes (PL), there is no Pauli blocking, which lies at the basis of the second contribution (shakeup). Therefore, the emission energy of indistinguishable complexes more closely resembles $\hbar\omega = E_{oc}$, which is redshifted with respect to the dR signal. The lack of an additional blue shifting term is also evidenced by the almost identical slope that the H-, O-, and M excitons show in PL. In addition, the photo-excited electron has more freedom to reside somewhere within the Fermi sea, which results in an asymmetric broadening towards higher energies, which increases with a rising Fermi level.

To address this fine comment, we have:

- Completely rewritten the paragraph that explains the observed PL emission as shown in Figure 4 of the main text. Starting from line 511.

(1) I generally agree with the experimental observations presented. However, I am concerned about the inconsistency between the dR and PL spectra regarding the Landau level resonances. The manuscript suggests that the presence of additional electronic reservoirs may explain the discrepancy. I think this could be plausible if the Landau level resonances appear in the dR spectra but are absent in PL. However, the opposite scenario is observed here so this casts doubt on the proposed interpretation of the M-exciton. The authors should clarify this inconsistency and provide a more robust explanation.

We thank the reviewer for this question, it is certainly on point and touches upon the limits of our understanding. In evaluating this question, we now believe that we have a better understanding of the discrepancy between dR and PL. However, we would like to point out that the interpretation of PL spectra is much more complicated than the interpretation of absorption spectra, and we do not yet understand every aspect of it.

We will answer this question by considering the observed LLs for different complexes. The hexciton as observed in dR in Fig 3 shows a fan of LLs. That is because, for absorption, many LLs are available in the upper K/K' valleys for the photoexcited electron-hole pair to be excited to. On the contrary, for PL measurements (Extended data Fig 4), all photoexcited electrons have time to relax to the lowest LL in the upper K/K' valley before recombination, and we observe no LLs for the hexciton. Then, when we shift to the oxciton, the dR data shows us a fan of LLs with progressively disappearing LLs for increasing gate voltage, this is due to LLs being filled and therefore becoming Pauli blocked. For PL, the opposite happens; we progressively see LLs appear, building up a fan of LLs. When an LL is filled, the photo-excited hole has the opportunity to form an excitonic resonance with a resident electron from this LL and recombine with it.

Extrapolating from these observations, we phenomenologically conclude that dR is sensitive to Landau levels in the upper K/K' valleys above the Fermi level, and PL is sensitive to Landau levels in the upper K/K' valleys below the Fermi level. To make this point more insightful, we have overlaid the peak-detection data of Fig. 3c in the main text with the PL data shown in Extended data Fig. 5c, shown in Fig. 1 of this resubmission.

When the Fermi level reaches the Q/Q' valleys, we observe the *M* exciton. A photoexcited electron that resides in an LL that is energetically above the Q/Q' valley can scatter between the Q/Q' - and the K/K' valleys. We argue that this availability of additional reservoirs (i.e. the Q/Q' valleys) causes dephasing and therefore broadening, such that the individual LLs are not resolvable anymore. This explains why no LLs are observed in dR. For PL, which is

sensitive to LLs below the Fermi level, the majority of the filled LLs in the K/K' valleys lie energetically below the onset of the Q/Q' valleys, and thus, electrons residing within them cannot scatter to the Q valley, and no broadening is observed. This understanding is further supported by the fact that when the signal in PL changes from the O- to the M exciton, no additional LLs appear. We are aware that for the M exciton in PL, as shown in Fig. 1 of this resubmission, the LLs first fade with increasing charge density and then become very pronounced again. At this point, we do not have a full understanding of this behavior. This will remain an open question for future work, and we believe that our results may stimulate theoretical efforts directed at obtaining a fuller understanding. Despite this discrepancy, we believe that observing LLs in PL but not in dR is consistent with our qualitative explanations.

In order to explain the discrepancy between the Landau levels observed in PL and dR in the manuscript we have:

- Revised the paragraph explaining the Landau levels observed for the M exciton in lines 462 to line 479.
- Rewritten the caption of Extended data Figure 5, including an explanation of our phenomenological observations.
- Extended the range of the plotted PL spectra in Fig. 5 to include more data related to the H exciton.

Figure 1. (a) The derivative of the electron density-dependent photoluminescence spectra at a magnetic field of 8 T resolved for σ^+ polarization, as in extended data Fig. 5c. Overlaid are the first five fitted resonances of reflection contrast measurements under the same experimental

conditions, taken from Fig. 3c in the main text. The electron density of both datasets has been calibrated using the LL resonances of the exciton as described in the methods of the manuscript. We clearly observe that the LLs observed for dR lie energetically below the levels observed for PL, and that in the case of the exciton, the resonances for dR continue into the resonances for PL. **(b)** A schematic representation of PL emission for the σ^+ polarization for our phenomenological model in analogy to Fig. 3d in the main text. In regime I and II, no LLs in the upper K valley are filled and we observe no LLs in the emission. Then, for regime III, we start filling increasingly more LLs and observe that the corresponding resonances appear. For regime IV we observe no additional resonances being created because LLs that lie above the onset of the Q valley suffer from scattering processes between the K and the Q valley.

Reviewer #2 (Remarks to the Author):

The authors performed reflectivity and photoluminescence measurements on a gate-tunable device based on WSe₂ (and WS₂) in a previously unexplored range of electron densities. By increasing the carrier concentration they tuned the Fermi level to observe many-body excitations (such as hexcitons and oxcitons) formed by charge carriers populating the higher-energy band with a minimum at the K/K' points. With further increases in electron concentration the authors accessed the regime of many-body complexes (M), which emerge as the Fermi level enters the Q/Q' valleys. In the latter part of the manuscript the authors investigate the properties of these M-complexes using optical spectroscopy performed in an external magnetic field and as a function of sample temperature.

I find this work very interesting and well-written, with a clearly conveyed message. The article is well-structured, and the presentation is appealing.

We thank the Reviewer 2 for these supportive words.

Below, I outline minor comments that the authors should address before publication in Nature Communications:

1. In the Introduction (paragraph starting at line 75), the authors discuss the trion/polaron controversy. However, this paragraph does not seem clearly connected to the rest of the manuscript. The trion/polaron issue is scarcely addressed in the main text, which focuses primarily on H, O, and M excitations. Clarifying the relevance of this discussion to the overall study would be beneficial. Otherwise, I suggest removing said paragraph.

We thank Referee 2 for this comment and agree that the relevance of this paragraph is ambiguous. We believe that finding a comprehensive model to understand all excitonic resonances will benefit from more experimental observations that can support or falsify their correctness. The measurement of the *M* exciton is therefore clearly relevant in this context. However, providing a comprehensive discussion of different theoretical models is out of the scope of this manuscript, where we focus on the experimental observations and explain them to the best of our understanding. We therefore fully agree with Reviewer 2 that the mentioned paragraph is best omitted to avoid confusion on the contents of the manuscript.

Please note that Referee 3 asked us to provide a more "explicit discussion of competing theoretical frameworks to strengthen the conclusion". Rather than extending the length of the manuscript with discussions of the various theoretical frameworks, we have decided to follow the recommendation of Referee 2 to shorten the discussion in the introduction. Our motivation is that this work provides sufficient compelling evidence to support our conclusions, and that a discussion on theoretical frameworks remains outside the scope of this work.

In order to reduce the discussion to its essentials, we have:

- Removed the said paragraph from the main text.

2. In the experimental section (starting at line 97), the temperature at which the measurements were conducted is not reported. Currently, the reader must refer to the Methods section to find this information. It would improve clarity to include the measurement temperature either directly in the main text or in the relevant figure caption.

We thank the Referee for pointing out that we did not mention the temperatures in the main text. This is indeed highly relevant information for interpreting the data, and we have now included it accordingly.

We have added the sample temperature of the corresponding measurements at the following points in the text:

- The caption of figure 1
- The caption of figure 2
- The caption of figure 3
- The caption of figure 4
- The caption of figure S1
- The caption of figure S5

3. In Figure 3, the tick positions differ between panels (a) and (b), yet panel (b) lacks tick labels. For consistency and readability, tick labels should be included in panel (b).

We acknowledge that the ticks used in figures 3 (a), (b) and (c) can cause confusion, and we thank the reviewer for pointing this out. The ticks on the left-hand side of each panel correspond to the voltage scale on the far left, and ticks on the right-hand side of each panel correspond to the density scale on the far right, but the ticks in each panel are identical. Because this figure is already packed with a lot of information, we chose not to add more tick labels, as this would negatively impact the readability of the figure overall. Instead, we improved the clarity by adding an explanatory sentence to the caption. In addition, we made the ticks on the right-hand side of each panel thicker to mark that they correspond to the labels on the far right hand side.

To clarify the ticks in panels (a), (b), and (c) we have:

- Added the sentence “Left (right) ticks in panels (a), (b), and (c) correspond to the voltage (density) scale on the far left (right) of the figure.” to the caption of figure 3.
- Made the ticks on the right-hand side of each panel thicker to signal that they correspond to the density scale on the far right of the figure.

4. Starting at line 424, the authors discuss a change of slope; however, this change is not easily discernible in the presented data. Including a guide to the eye would help the reader follow this discussion more easily.

We thank the reviewer for this suggestion to improve the presentation of the data. The change in slope is indeed not easily discernible, and we have included a guide to the eye in Figure 3.

To improve the presentation of the data we have :

- Included a guide to the eye in figure 3
- Updated the caption of figure 3
- Updated the sentence on line 425, which now reads: “A striking feature of the transition from regime I to II in Fig. 3b is the change in slope of the H resonances, marked with a black arrow, which signals a change in the rate at which the Fermi level rises under the injection of carriers.”

As mentioned above, the paper is well-written and, in my opinion, complete. However, I would like to offer a few observations that could further improve the manuscript. I emphasize that it is not necessary for the authors to follow any of the suggestions below for successful publication in Nature Communications.

We thank the reviewer for the suggestions below. We also want to express our appreciation for the choice to clearly separate comments that need to be addressed for publication from suggestions that can improve the overall manuscript.

1. Is the oscillator strength (OS) conserved when the O excitation transitions to the M excitation? Have the authors determined the OS from the reflectivity (R) spectra?

Reviewer 2 raises a very interesting point, as the oscillator strength can tell us more about the nature of the resonance. To make this value quantitative, we have performed dispersive Lorentzian fits of the *H*, *O* and *M* resonances. These fits provide values for the amplitude and the width of the resonances. What we clearly observe is that the amplitude does not significantly decay when switching from *H* to *O*, nor from *O* to *M*. On the contrary, we do clearly observe that the *O* and *M* excitons broaden as the electron density increases, in addition to a step in the width when transitioning from *H* to *O* and *O* to *M*. These observations agree with the composite exciton model that we use to explain our data.

To include insights into the oscillator strength in the manuscript, we have:

- Added extended data figure 2, in which dispersive Lorentzian fits of the reflection contrast data are shown together with the fitting results on center energy, amplitude, and width for the *H*, *O* and *M* excitons.
- Added a section to the methods labelled 'Reflection contrast fitting' starting on line 660 explaining the fitting procedure.
- Added a sentence to the main text on line 229 that reads:
"This transition does not lead to a loss in oscillator strength, which remains almost constant for the *H*, *O*, and *M* resonances (see Extended data Fig. 2)."

2. Line 510 – I believe the strong photoluminescence (PL) observed for the O and M excitations is not surprising, as the PL yield is a product of oscillator strength and occupation. The occupation should be high since O and M correspond to the energetically lowest excitations.

The Referee raises a very valid argument for the PL observed for *O* and *M* being bright. Adding to these valid reasons, other factors can be as crucial as the one mentioned by the Referee. For example, the laser energy is about 300 meV larger than the energy of the emitted photons from *H*, *O* and *M*, which means that the time it takes to form a thermalized complex is shorter when the electron density increases because there are more resident electrons with which the photoexcited electron-hole pair can scatter. Faster formation of a thermalised complex also means that non-radiative channels play a lesser role when we enter the *O* and *M* regimes. Furthermore, the density of created photoexcited pairs can be different in each of the regimes because new ways become available to form a bound complex when more valleys are populated (while preserving momentum and energy). Since elaborating on the relaxation and formation pathways is a work of its own, we opted not to elaborate on this point in this work.

3. The lack of a blueshift for the O transition in PL should be addressed—some hypothesis or discussion would be helpful.

This is a very valid point that was also pointed out by Reviewer 1. We discuss the difference in mechanism for RC and PL in the answer to question 2 of reviewer 1 and explain why this results in the lack of an observed blueshift for the exciton in PL.

In order to explain this discrepancy between PL and RC in the manuscript, we have:

- Completely rewritten the paragraph that explains the observed PL emission as shown in Figure 4 of the main text. Starting from line 511.

4. Line 466 – Landau levels (LLs) are not visible in reflectivity but are clearly observed in PL for the M excitation. Could the visibility of LLs in PL provide insights into the scattering rate?

We thank the Reviewer for raising this very relevant point. This question is closely related to question 2 of Reviewer 1. We would like to refer to our answer to this question for an explanation of our best understanding of why there are no LLs visible in reflectivity, but there are in PL. Our best understanding is indeed based on a scattering process between the K- and the Q valley. However, this explanation is somewhat speculative, touches upon the limits of our current understanding, and more experimental evidence is required to truly claim that this mechanism is responsible. We therefore choose not to attempt to estimate the scattering rate and leave this as an open question for future work.

5. Line 399 – The increase in the splitting between consecutive LLs is most likely due to band non-parabolicity. A brief discussion or acknowledgment of this point could add value.

Referee 2 touches upon a very interesting point that appears in our dataset, and we agree that we do not provide an explanation for the observed increase in LL spacing. A consecutive splitting between Landau levels can indeed be well explained with band non-parabolicity, but this is not what we observe. In our data, the spacing between Landau levels at a given voltage is constant, and it is this constant spacing that changes when increasing the gate voltage. We speculate that the observed increase in LL spacing with voltage may be due to an enhancement in binding of the photoexcited electron to the complex.

It was not very clear from the main text that the spacing between LLs remains constant for a given voltage, and we have altered the sentence explaining this. In addition, we have added a comment with our best understanding to explain the increasing spacing.

To improve our discussion on the increase in the splitting between consecutive LLs we have:

- Changed the sentence on Line 394 which now reads:
“We start from the energetic spacing of the resonances in regime I (ΔLL), which is constant for a given voltage but increases slightly from 3.2 meV to 3.6 meV from the bottom to the top of regime I and is identical for both polarizations.”
- Added a comment to line 409 which reads:
“The slight change of the energetic spacing of the resonances might be due to an enhancement in binding of the photoexcited electron to the complex[9].”

Reviewer #3 (Remarks to the Author):

This manuscript by Dijkstra, et al. reports the experimental discovery of a previously unexplored many-body excitonic complex (“M”) in electrostatically doped monolayer WSe₂. Using 5-6 nm thin hBN gate dielectrics to reach exceptional high charge densities, the authors investigate the formation of neutral, charged, and many-body excitonic complexes via reflection contrast and magneto-optical spectroscopy. A key result is the identification of a thermodynamically stable exciton involving interactions with up to 10 filled conduction band valleys, potentially comprising as many as 20 quasiparticles—a remarkable extension of known excitonic states. The finding provides a unique insight into extremal many-body physics in two-dimensional semiconductors and opening pathways for testing the limits of excitonic models under strong doping.

Overall, the data is of high quality, and the result contributes a great new insight to quantum many-body excitonic complexes in 2D materials with broad impact to condensed matter research and quantum physics. The text is well organized and clearly written. I recommend the publication of this manuscript in its present form.

We thank the reviewer for their kind words.

Major Comments:

While the manuscript discusses the “M” complex as involving interactions with up to 9 Fermi particle-hole excitations (yielding up to 20 total particles), it remains somewhat speculative in the absence of direct spectroscopic signatures of individual components.

This is a prudent remark and a very layered question.

The Reviewer is correct that we do not observe a clear spectroscopic feature, such as an energetic step, that could be the hallmark of yet another Fermi electron-hole pair binding to the overall complex and give a clue to its exact size. At the same time, this is not surprising because the observed binding energies are increasingly smaller for each additional Fermi electron-hole pair that binds to the complex. For example, the binding energies of the X_T^- , X_S^- , H , and O excitons are ~27 meV, ~34 meV, ~10 meV and ~2 meV respectively. We thus cannot tell the difference between a complex with say 7 or 9 particle-hole excitations. Our argument for why we expect all ten valleys do participate is that the photoexcited (or recombining) e-h pair cannot be selective with which of the electrons in its vicinity it interacts. For example, there are 10 electrons within a radius of 8 nm away from the photoexcited electron-hole pair when the electron density is $1.5 \cdot 10^{13} \text{ cm}^{-2}$. If these electrons are distinguishable in their spin and valley, they can completely overlap in space without violating the Pauli exclusion principle, reinforcing the understanding that the photoexcited electron-hole pair interacts with them concurrently.

The manuscript would benefit from explicitly addressing this limitation and delineating the extent to which the proposed configuration is supported by experimental observables (e.g., energy shifts, broadening, thermal stability) versus modeled assumptions.

We agree with the reviewer that a more quantitative analysis will strengthen the message. Therefore, we fitted the H , O and M excitons of our reflection contrast data with a dispersive Lorentzian function. What we find is that we observe no significant change in the amplitude (i.e. oscillator strength) for any of the three resonances. Contrastingly, the full width at half maximum shows an almost constant width for the H exciton, which agrees with the composite excitonic states model, because it is an optimal complex with a distinguishable photoexcited electron-hole pair. Then, starting from the transition from H to O , we observe a continuous broadening that also extends to the M exciton. These observations are expected

for O and M , as they are complexes with an indistinguishable photoexcited electron-hole pair. In addition, there are steps in the width of the resonance when transitioning from H to O and from O to M , which clearly mark the onset of a new type of resonance. Qualitatively, all these observables are in clear agreement with our model.

To provide more observables that support our theoretical model we have:

- Added extended data figure 2, in which dispersive Lorentzian fits of the reflection contrast data are shown together with the fitting results on center energy, amplitude, and width for the H , O and M excitons.
- Added a section to the methods labelled 'Reflection contrast fitting' explaining the fitting procedure.
- Added a sentence to the main text on line 222 that reads:
"The shakeup also causes the resonance to broaden with increasing charge density, confirmed by dispersive Lorentzian fits of the reflection contrast spectra shown in Extended data Fig. 2."
- Rewrote two sentences in the main text on line 226 that now read:
"Finally, $V_G \sim 4.2 V$ (purple arrow in Fig. 1b) marks yet another transition---this time from O to M , with a change from blueshifting to redshifting behavior and a sudden increase in width. This transition does not lead to a loss in oscillator strength, which remains almost constant for the H , O , and M resonances (see Extended data Fig. 2)."

To quantitatively compare these observables to our model is challenging. Currently, we cannot use our composite excitonic complex formalism to compute the binding energy of complexes beyond the hexciton. The computational effort to calculate the binding energy of 8-body complexes is beyond the reach of our supercomputer, let alone 20-body. For this work, we can therefore not provide a direct quantitative comparison with our model.

More explicit discussion of its competing theoretical frameworks would strengthen the conclusion.

We thank the Reviewer for this comment, and we agree that a clear and explicit discussion of competing theoretical frameworks would be beneficial for the field as is. We would like to note that Reviewer 2 suggested removing a paragraph in the introduction that mentions competing theories, as it did not reflect the content of the paper, and we agree. The main point of this work is to share the experimental observation of a new resonance that we explain to the best of our knowledge. We would not do it justice by also attempting to provide a comprehensive discussion of different theoretical models.

In order to leave the focus on the experimental observation, we have:

- Removed the paragraph from line 75.